# Metabolomic Profiling Reveals Sex Specific Associations with Chronic Obstructive Pulmonary Disease and Emphysema

**DOI:** 10.3390/metabo11030161

**Published:** 2021-03-11

**Authors:** Lucas A. Gillenwater, Katerina J. Kechris, Katherine A. Pratte, Nichole Reisdorph, Irina Petrache, Wassim W. Labaki, Wanda O’Neal, Jerry A. Krishnan, Victor E. Ortega, Dawn L. DeMeo, Russell P. Bowler

**Affiliations:** 1Computational Bioscience Program, University of Colorado Anschutz Medical Campus, Aurora, CO 80045, USA; lagillenwater@gmail.com; 2Department of Biostatistics and Informatics, Colorado School of Public Health, University of Colorado Anschutz Medical Campus, Aurora, CO 80045, USA; KATERINA.KECHRIS@CUANSCHUTZ.EDU; 3Division of Medicine, National Jewish Health, Denver, CO 80206, USA; prattek@njhealth.org (K.A.P.); PetracheI@njhealth.org (I.P.); bowlerr@njhealth.org (R.P.B.); 4Skaggs School of Pharmacy and Pharmaceutical Sciences, University of Colorado Anschutz Medical Campus, Aurora, CO 80045, USA; NICHOLE.REISDORPH@CUANSCHUTZ.EDU; 5School of Medicine, University of Colorado, Aurora, CO 80045, USA; 6Division of Pulmonary and Critical Care Medicine, University of Michigan, Ann Arbor, MI 48109, USA; wlabaki@med.umich.edu; 7Cystic Fibrosis/Pulmonary Research and Treatment Center, University of North Carolina at Chapel Hill, Chapel Hill, NC 27599, USA; 8Breathe Chicago Center, University of Illinois at Chicago, Chicago, IL 60608, USA; jakris@uic.edu; 9Center for Precision Medicine, Department of Internal Medicine, Wake Forest School of Medicine, Winston-Salem, NC 27157, USA; vortega@wakehealth.edu; 10Channing Division of Network Medicine, and Division of Pulmonary and Critical Care Medicine, Department of Medicine, Brigham and Women’s Hospital, Harvard Medical School, Boston, MA 02115, USA

**Keywords:** COPD, emphysema, sex differences, network analysis, weighted gene co-expression network analysis (WGCNA), lung

## Abstract

Susceptibility and progression of lung disease, as well as response to treatment, often differ by sex, yet the metabolic mechanisms driving these sex-specific differences are still poorly understood. Women with chronic obstructive pulmonary disease (COPD) have less emphysema and more small airway disease on average than men, though these differences become less pronounced with more severe airflow limitation. While small studies of targeted metabolites have identified compounds differing by sex and COPD status, the sex-specific effect of COPD on systemic metabolism has yet to be interrogated. Significant sex differences were observed in 9 of the 11 modules identified in COPDGene. Sex-specific associations by COPD status and emphysema were observed in 3 modules for each phenotype. Sex stratified individual metabolite associations with COPD demonstrated male-specific associations in sphingomyelins and female-specific associations in acyl carnitines and phosphatidylethanolamines. There was high preservation of module assignments in SPIROMICS (SubPopulations and InteRmediate Outcome Measures In COPD Study) and similar female-specific shift in acyl carnitines. Several COPD associated metabolites differed by sex. Acyl carnitines and sphingomyelins demonstrate sex-specific abundances and may represent important metabolic signatures of sex differences in COPD. Accurately characterizing the sex-specific molecular differences in COPD is vital for personalized diagnostics and therapeutics.

## 1. Introduction

Susceptibility to and progression of lung disease, as well as response to treatment, often differ by sex, but molecular pathways underlying these differences are poorly understood [1,2]. Chronic obstructive pulmonary disease (COPD) is a progressive lung disease characterized by persistent airflow limitation associated with chronic inflammation [3]. As of 2017, chronic lower respiratory diseases, including COPD, ranked as the 3rd leading cause of death in females and the 4th leading cause of death in males in the US, accounting for 6.2% and 5.2% of deaths, respectively [4]. There are several reasons to consider analyzing the metabolome of COPD separately in men and women, including sex differences in age of onset of severe COPD [5,6,7], prevalence of airway disease and emphysema [8,9], and COPD-related comorbidities [10]. There have been reports of metabolome differences by sex [11,12,13]. Several classes of metabolites, including sphingomyelins and branched chain amino acids, have been associated with COPD phenotypes [14,15,16,17]. However, these studies were often small, used targeted metabolomics, and did not elucidate the sex-specific differences for COPD and emphysema.

Untargeted metabolomics identifies many molecules present with relatively low molecular weights (<1500 Da) that can represent functional endpoints of cellular regulation, exogenous exposures (e.g., food, drugs, tobacco smoke), and pathophysiological conditions [18]. Metabolomic profiles can be obtained from any biologic tissue or biofluid, with the metabolomic profile of COPD having previously been investigated in blood, urine, breath condensate, and bronchial lavage fluid [19,20,21]. Blood (plasma and serum) is often preferred for biomarker discovery because it is a minimally-invasive biosample and blood sampling is widely available in many settings. Furthermore, although the primary target organ of COPD is the lung, COPD is a systemic disease with blood signatures [22,23,24].

Sex-specific differences are not randomly distributed over the metabolome, but likely are manifest in co-regulated metabolic pathways [11,13]. Weighted gene co-expression network analysis (WGCNA) is an approach developed for hierarchical clustering of gene expression data into modules of correlated genes which can be extended to metabolomics [21,22,23,25,26]. In identifying modules of correlated metabolite abundances, one can explore subnetworks that are differentially dysregulated within COPD subjects by sex and identify sex-specific biomarkers of those subnetworks [27]. Until recently, there have not been large COPD cohorts with metabolomics data that could be used to assess sex-specific metabolomic variability in COPD. In this study, we hypothesized that there are sex differences in metabolomic modules that associate with COPD status and emphysema. To test this hypothesis, we utilized an untargeted metabolomics discovery platform in a well-characterized COPD cohort (COPDGene), as well as a separate well-characterized independent COPD cohort, SubPopulations and InteRmediate Outcome Measures In COPD Study (SPIROMICS).

## 2. Results

### 2.1. Demographics

In the COPDGene cohort, males were significantly older, had more smoking pack-years (though fewer current smokers), had a higher percentage of COPD cases, and a higher mean of percent emphysema than females (Table 1). Similar sex-specific differences were observed for smoking pack-years and COPD cases in SPIROMICS. Overall, the COPDGene subjects were older, contained a smaller percentage of African American subjects, had a lower percentage of current smokers, had less smoke exposure measured by cigarette pack-years, and less percent emphysema compared to SPIROMICS subjects (Appendix A).

### 2.2. WGCNA Modules

For COPDGene, a soft-thresholding power (β) of 8 was chosen, based on criteria for scale-free topology within a signed network (see Appendix A) [28]. Following the WGCNA procedure outlined in the methods, 11 modules of co-varying metabolites were identified (not including the “grey” module of uncorrelated metabolites) (Figure 1A). Modules largely segregated based on metabolite sub class (Table 2, full module assignment in Appendix A). Of the clinical variables tested for univariate associations, sex was significantly correlated with the most modules, followed by age (Appendix A). Significant associations were observed between individual models and each clinical variable tested, with several modules significantly correlated to more than one variable. Thus, covariate adjusted models were necessary to identify independent associations.

For SPIROMICS, a soft-thresholding power (β) of 9 was chosen and 11 modules of co-varying metabolites were identified (Figure 1B). As in COPDGene, modules largely segregated based on metabolite sub class and were most significantly associated with sex and age (Table 2, Appendix A).

Module preservation was observed between COPDGene and SPIROMICS, with all but the pink and purple modules in COPDGene having a corresponding module in SPIROMICS (Figure 1C, Appendix A). The preservation of the 9 modules is further supported by the overlap of significant hub metabolites between cohorts (Table 3). While we explored consensus clusters between COPDGene and SPIROMICS, we decided to use clusters found from the individual clusters based on the high preservations and difficulty in harmonization between platforms (Figure 1C).

In the sex-stratified analysis of COPDGene, 13 and 15 modules were found for the metabolomic data from females and males, respectively. In comparing modules between sexes, a corresponding module in the opposite sex was observed for almost all modules, though for female modules blue, black, brown, turquoise, and red there were there were groups of metabolites split over the male modules (Figure 2). We then compared module preservation between each sex and the full cohort. Here we found the female modules almost exactly corresponding with the modules of the full cohort, while some of the full cohort modules were split over individual male modules (Appendix A). Overall, high module correspondence was found between module assignments in each sex and the module assignments from the full cohort, thus full cohort module assignments were used in association testing in the stratified models. In comparing sex-specific modules in SPIROMICS, preservation of modules structure is observed between sexes, though, like COPDGene, there are some cases where a module in one sex is split over two modules in the other (Appendix A).

### 2.3. Covariate Adjusted Module-Phenotype Associations in COPDGene

Modules were summarized for phenotype association analysis using the eigenvalue. This summarizes the profiles the metabolites within the module into a single orthogonal vector summarizing the majority of the variance within the module. Thus, the noise of individual metabolite variances is reduced to better summarize metabolic function over highly correlated metabolite classes.

#### 2.3.1. Sex

In COPDGene, significant differences by sex were observed in 9 of the 11 modules (Figure 1D, Appendix A). The most significant differences were observed in the red module, for which metabolite abundances were higher in females, and the magenta module, for which metabolite abundances were higher in males. In all, the red, green, blue, black, and greenyellow modules had higher abundances in females, while the turquoise, magenta, pink, and brown modules had higher abundances in males. Similarly, 8 of the 11 SPIROMICS modules significantly differed by sex (Figure 1E, Appendix A).

#### 2.3.2. COPD

For COPD case status in COPDGene, significant associations were observed in 5 of the 11 modules for the full cohort (Figure 1D). The most significantly associated modules were the black and brown modules, with lower metabolite abundances observed in COPD cases. The black and brown modules were also associated with increased BMI over the full cohort and within each sex. While the yellow module did not significantly differ by sex, the metabolite abundances were significantly lower in COPD cases in the full cohort and lower in African Americans. In SPIROMICS, at a nominal *p* value of 0.5, the the magenta module was negatively associated with COPD (Figure 1E).

In the sex-stratified analyses, the red, pink, and brown modules were significantly associated in the male strata in COPDGene. Of note, higher values in the red module were associated with COPD status in the male stratum, though not in females or the full cohort (Figure 1D). In SPIROMICS, a nominal relationship was observed between the magenta module and COPD within the male stratum (Figure 1E).

#### 2.3.3. Percent Emphysema

For COPDGene, the brown, magenta, and turquoise modules were all associated with percent emphysema (Figure 1D). In the sex stratified analyses, the brown and magenta modules were also associated with percent emphysema in males, while the turquoise module was significant in females. In SPIROMICS, only one female specific nominal association was observed between the green module and emphysema (Figure 1D).

#### 2.3.4. Covariates

We also tested the other clinical variables, serving as covariates in the COPD phenotype models, as primary predictors in multiple regression models. In COPDGene, age and BMI were significantly associated with the most modules (7 and 6, respectively) over the full cohort, as well as within the sex stratified analyses (Figure 1D). Of note for age associations are the brown and black modules, which were not associated over the full cohort but only associated in females and males, respectively. For BMI, the pink and magenta modules were only significant in the female stratum. Race was most strongly associated with the yellow module, though significant associations were also seen with the red and greenyellow modules. The greenyellow module was also significantly associated with current smoking status and smoking pack-years. The magenta module was further associated with current smoking status in the full cohort and in males, while the brown and black modules were significantly associated with smoking in males and females, respectively.

In SPIROMICS, similar relationships were observed among many of the preserved modules and the covariates tested, though not all (Figure 1D,E). For example, associations were not observed between the SPIROMICS green module (preserving the COPDGene green module) and age. Also, the SPIROMICS black module (preserving COPDGene yellow) was associated with smoking status and intensity, while the magenta module (preserving the COPDGene greenyellow) was associated with smoking intensity.

### 2.4. Individual Associations

#### 2.4.1. COPD Modules

Within modules significantly associated with sex and COPD in COPDGene (black, blue, brown, pink, and red), 88 of the 501 module metabolites were significantly associated with COPD in the full cohort, 24 of which were significant in females and 27 of which were significant in males (Appendix A) (multiple comparison corrections were made over the entire metabolome). Three metabolites (ceramide (d18:1/17:0, d17:1/18:0)*, octadecenedioate (C18:1-DC)*, and N-stearoyl-sphingosine (d18:1/18:0)*) were significant only in the male stratum (not also in the full cohort). In the sex-specific associations, 8 metabolites were significant in both sexes including inverse associations with retinol (Vitamin A), phosphocholine, and xenobiotics ergothionene and 3-formylindole, and positive associations with 4 acyl carnitines (Figure 3). The most represented sub pathway male-specific COPD associations were sphingomyelins (7/20), while phosphatidylethanolamines and acyl carnitines were most represented in females with COPD (4/20 for both sub pathways). There were several metabolites in which opposite directions of associations were observed between sexes, though only ceramide (d18:1/17:0, d17:1/18:0)* was significant in either sex (males) (Figure 4).

Using the SPIROMICS modules preserving the COPDGene sex and COPD-associated modules, we performed a bi-directional lookup between cohorts. The only significant sex specific associations were between hypotaurine and beta-cryptoxanthin in males, though many metabolites were nominally significant (*p* < 0.05, before multiple comparison correction) (Appendix A). Among the metabolites nominally associated in SPIROMICS, the male specific association with succinate was replicated between cohorts, as well as female-specific associations with acyl carnitines.

#### 2.4.2. Percent Emphysema

Within modules significantly associated with sex and percent emphysema (brown, magenta, and turquoise), 3 metabolites were significantly associated with percent emphysema in the full cohort (5-hydroxylysine, isovalerate (C5), X-17357), none of which were significant in females and 2 of which were significant in males (5-hydroxylysine, X-17357) (Appendix A). Three metabolites that were not statistically significant in the full cohort were significant in the female sex-specific analysis (2,3-dihydroxy-2-methylbutyrate, alpha-ketoglutaramate*, and homocitrulline) (Figure 5).

## 3. Discussion

This is the largest metabolomic analysis of COPD cases to date and the first to examine associations for sex-specific modules associated with COPD and emphysema. While many of the metabolite modules were similar for men and women, we identified several modules of correlated metabolites in COPDGene that significantly differed by sex, COPD status, and percent emphysema. In sex-stratified analyses, we further identified associations specific to, or largely driven, by one sex. In particular, acylcarnitines and phosphatidylethanolamines (PEs) were significantly greater in females with COPD, while sphingomyelins were greater in males with COPD. While the general direction of these associations was similar between cohorts, not all COPD sex-specific metabolite associations found in COPDGene were statistically significant in SPIROMICS. However, multiple acylcarnitines had significant sex-specific associations in both cohorts. These findings are further evidence of sex differences in molecular dysregulation during COPD pathogenesis that need to be considered in study design and personalized treatment development.

The metabolomic profiles strongly differed by sex, with the most significant differences found in separate modules of sphingolipids and steroids. Over all subjects with and without COPD, sphingolipid metabolite abundances were higher in females, confirming previous observations [11,29]. We further replicated the observed sex-specific discordance in sphingolipids by age (increasing in females and decreasing in males) from a recent longitudinal analysis of 1212 participants in the Wisconsin Registry of Alzheimer’s Prevention [12]. The higher levels of androgenic, pregnenolone, and progestin steroids in men confirmed previous findings [29,30], as were the decreases in both men and women by age [12].

The lipid steroid module was also higher in current smokers overall and within the male stratum. This replicates findings of higher androgens abundances in males though does not reveal elevated levels of androgenes observed in both pre- and post-menopausal women [31,32,33,34]. The module including cofactors and vitamins involved in ascorbate/aldarate (i.e., oxalate) metabolism among other sub pathways, was significantly higher in females and lower among current smokers. This confirms previous observations of lower metabolite abundances in those pathways among cigarette smokers, but also adds a novel sex specific association for future research [34,35].

The sex-specific associations for acylcarnitines and COPD may be related to mitochondrial dysfunction. Higher circulating acylcarnitne abundances with increased inflammation and risk of cardiovascular disease [34,35,36]. Physiologically, acyl carnitines function in mitochondrial metabolism of fatty acids, the removal of excess acyl groups from the body, and the modulation of intracellular coenzyme A homeostasis [37]. While acyl carnitines dysregulation affects both sexes, in COPD the effect is more pronounced in females [38,39,40,41,42,43]. Recently, another analysis within SPIROMICS identified positive correlations between urine mitochodrial DNA (mtDNA), an indicator of mitochondrial dysfunction, and respiratory symptoms specifically within females [44]. Since mtDNA is maternally inherited, our findings combined with the previous literature support the hypothesis of heritable sex differences in mitochondrial regulation leading female-specific subphenotype within COPD [45].

Sphingomyelins and other plasma membrane-complex lipid molecules, including phosphatidylethanolamines and phosphatidylcholines, have been observed at higher abundance in females [12]. Sphingomyelins have been previously associated with COPD in the plasma of subjects enrolled in COPDGene, assessed using targeted metabolomics [14]. Higher sphingomyelin abundances among COPD subjects were observed in both sexes in COPDGene, though the difference was only significant among males and this sex-specific shift was not observed in SPIROMICS. Phosphatidylethanolamines have also been associated with COPD phenotypes, though the direction of effect is not always consistent [27,42]. In this study, we observed lower values in COPD subjects for both sexes, though statistical significance was only observed among females. Phosphatidylethanolamines have been functionally associated with protein biogenesis, oxidative phosphorylation, autophagy, membrane fusion, mitochondrial stability, and act as precursors of other lipids [46].

This study has several limitations. Firstly, samples were obtained from plasma, a biofluid, which while representative of systemic metabolic regulation, may not fully capture the impact of the first line of exposure to tobacco smoke in the lung. A recent study of bronchoalveolar lavage fluid (BALF) by our group found much more robust metabolomic associations with COPD phenotypes in BALF compared with blood [21]. However, BALF collection is not without risk in individuals with advanced COPD, and is generally limited to collection is smaller numbers of subjects thus limiting application to the larger cohort sized needed to investigate sex-specific effects. Second, while the COPDGene and SPIROMICS cohorts are two large, well-characterized COPD cohorts, pharmaceutical data and diet differences were not considered. Third, 24% of the COPDGene metabolites were unannotated. These were kept in the analysis since Metabolon is consistently elucidating unannotated metabolites and may be helpful for future interpretation despite limiting current interpretation. Finally, the threshold of ≤20% missingness of metabolites removes many xenobiotics (such as cotinine) and other metabolites which may have significant associations among subjects in which they are present (e.g., sex-specific steroids).

## 4. Methods

### 4.1. Study Populations

COPDGene. The NIH sponsored multicenter Genetic Epidemiology of COPD (COPDGene) (ClinicalTrials.gov (accessed on 30 December 2020) Identifier: NCT00608764) study was approved and reviewed by the institutional review board at all participating centers [47]. All study participants provided written informed consent. This study enrolled 10,198 non-Hispanic white (NHW) and African American (AA) individuals from January 2008 until April 2011 (Phase 1) who were aged 45–80 with a ≥10 pack-year smoking history and no respiratory exacerbations for >30 days. From July 2013 to July 2017, 5697 subjects returned for an in-person 5-year follow up visit. Each in-person visit included spirometry before and after albuterol, quantitative CT imaging of the chest, and blood sampling. From two clinical centers (National Jewish Health and University of Iowa) 1136 subjects (1040 NHW, 96 AA) participated in an ancillary study in which they provided fresh frozen plasma collected using an 8.5 mL p100 tube (Becton Dickinson) at Phase 2. Never smokers, subjects classified as having mild COPD (forced expiratory volume at 1 s (FEV1) ≥ 80% predicted and FEV1/Forced Vital capacity (FVC) < 0.7) [48], and subjects with preserved ratio impaired spirometry, defined as a reduced FEV1 < 80% predicted with FEV1/FVC ≥ 0.7 [49] were excluded to define a final cohort of 839 subjects for analysis.

SPIROMICS. The Subpopulations and Intermediate Outcome Measures in COPD Study (SPIROMICS) (ClinicalTrials.gov Identifier: NCT01969344) includes 2771 subjects, aged 40–80 years with at least 20 pack-years of smoking. An additional 202 subjects were never smokers. Subjects had clinical phenotyping similar to COPDGene and fasting blood drawn at the enrollment visit using a p100 tube [50]. The first 649 subjects who returned for a 5–7 year visit (Visit 5) were selected for this study to match the 5-year follow up subjects from COPDGene; however, the blood profiled and phenotypes used were from the year 1 visit. After restricting to only include NHW and AA subjects, the same exclusion criteria were applied to SPIROMICS as COPDGene, leaving a sample size of 446 subjects for analysis.

### 4.2. Clinical Data and Definitions

COPD case status was defined using spirometric evidence of at least moderate airflow obstruction (the ratio of post-bronchodilator Forced Expiratory Volume at one second over Forced Vital Capacity FEV_1_/FVC < 0.50 and FEV_1_ percent predicted (FEV_1_pp) < 80% [51]. Control subjects were defined by an observed FEV_1_/FVC > 0.7 and a FEV_1_pp > 80%. Percent emphysema was quantified by percent of lung voxels less than 950 Hounsfield Units (% low attenuation area: %LAA) on the inspiratory CT scans. Visual emphysema was assessed as previously described for COPDGene [52,53] and SPIROMICS [54,55]. Subjects missing CT data (55 subjects in COPDGene and 63 in SPIROMICS) were excluded from emphysema analyses.

Plasma was profiled using the Metabolon Global Metabolomics Platform (Durham, NC, USA) [56,57,58]. The data were further normalized to remove batch effects, filtered by metabolites with high missingness, and imputed to remove missing values [59]. Metabolite “Super Class” and “Sub Class” annotations were determined by Metabolon (Durham, NC, USA).

### 4.3. Statistical Analysis

#### 4.3.1. Data Sets and Availability

Clinical data with definitions can be found on dbGaP for COPDGene (phs000179.v6.p2) and SPIROMICS (phs001119.v1.p1). For COPDGene and SPIROMICS, the following clinical data were used: COPDGene_P1P2_All_Visit_29Sep2018 and V5_DERV_INV1_200127, respectively. COPDGene and SPIROMICS metabolomic data are available at the NIH Common Fund’s National Metabolomics Data Repository (NMDR) website, the Metabolomics Workbench, https://www.metabolomicsworkbench.org (accessed on 30 December 2020) (Study IDs ST001443 and ST001639, respectively).

#### 4.3.2. Software

All analyses were run in R version 3.6.3 [60]. WGCNA.

The WGCNA R package was used to cluster groups of strongly related metabolites into networks [61,62]. The workflow of WGCNA included creating a signed matrix of Pearson correlations between metabolites, and transforming these into an adjacency matrix through soft thresholding by raising it to a power β based on the criteria for meeting scale-free topology (we chose to use Pearson correlation as opposed to Spearman or biweight midcorrelation based on previous inverse normal quantile transformation of the data).To identify this soft thresholding power, we tested a range of beta values between 1 and 30. The adjacency matrix was transformed into a topological overlap matrix (TOM). We used average linkage hierarchical clustering to group metabolites based on the topological overlap of their connectivity, followed by a dynamic tree-cut algorithm to cluster dendrogram branches into modules of highly correlated metabolites with a minimum module size of 15 metabolites [63]. For each metabolite, we calculated a Module Membership (kME) by correlating the metabolite values with the first principal component of the metabolite values in that module. “Hub” metabolites, which are considered to be central to the module, were defined as having a kME greater than 0.75 with their assigned module.

Pearson correlation between the first eigenvalue from each module and the clinical variables was used to determine the univariate significance of association between clinical variables and modules. To assess clinical associations independently, regressions were performed between the first eigenvalue of each model (dependent variable) and sex, COPD case status, percent emphysema, and other covariates including age, race, body mass index (BMI), current smoking status, smoking pack-years, and clinical center (each model was adjusted for all other covariates, with percent emphysema models being adjusted for FEV_1_pp as well) [64].

Module preservation between cohorts and sex-stratified populations was determined using one-sided Fisher tests of the observed overlap between modules [28,64].

#### 4.3.3. Individual Associations

Every metabolite was tested for associations with COPD case status and percent emphysema. Linear regression models were performed with the metabolite as the outcome and phenotype as predictor, adjusting for sex, age, race, body mass index (BMI), current smoking status, smoking pack-years, and clinical center, with percent emphysema models also being adjusted for FEV_1_pp. Sex-stratified models were also evaluated adjusting for all aforementioned covariates except sex.

Both the WGCNA and individual metabolite association analyses were performed on the data from the SPIROMICS cohort to carry out a bi-directional lookup between cohorts for sex-specific associations. WGCNA module membership was compared between COPDGene and SPIROMICS to compare module metabolite assignments. The SPIROMICS modules were assessed for associations with COPD status and percent emphysema as above.

## 5. Conclusions

Network analyses reveal important metabolic pathways associated with sex-specific features of COPD. While many modules had similar associations for both men and women, we identified sex-specific differences in acylcarnitines and several metabolites including succinate and ceramide (d18:1/17:0, d17:1/18:0)*. This suggests that metabolite profiles should be studied separately between men and women to understand sex-specific features of complex diseases. Moreover, as men and women may respond differently to treatments that target metabolism, there is a clear need for sex-specific metabo-therapeutics in the treatment of COPD.

## Figures and Tables

**Figure 1 metabolites-11-00161-f001:**
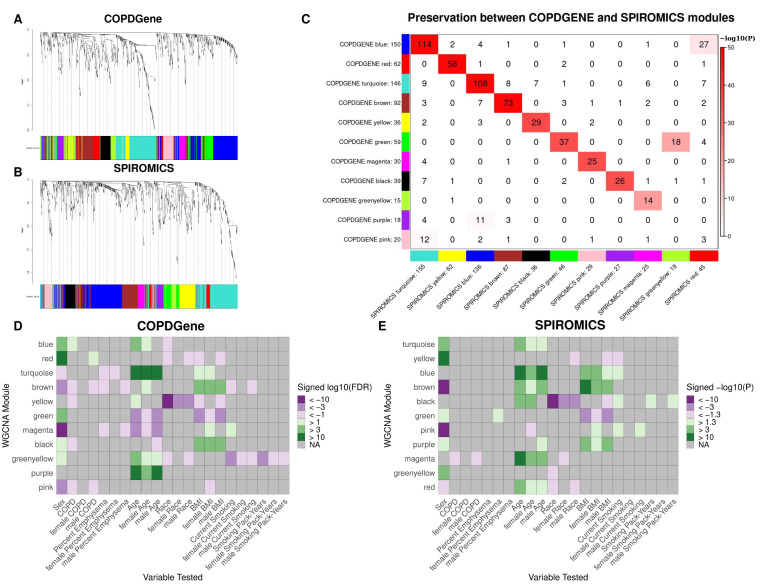
WGCNA results. (**A**,**B**). Hierarchical clustering tree (dendrogram) of genes based on human brain co-expression network for COPDGene (**A**) and SPIROMICS (**B**). Each “leaf” (short vertical line) corresponds to one gene. The color rows below the dendrogram indicate module membership. (**C**) Module Preservation between cohorts. Each row of the table corresponds to one SPIROMICS module (labeled by color as well as text), and each column corresponds to one COPDGene module. Numbers in the table indicate metabolite counts in the intersection of the corresponding modules. Coloring of the table encodes−log(*p*), with *p* being the Fisher’s exact test *p*-value for the overlap of the two modules. The darker the red color, the more significant the overlap is. D-E. Heat maps showing association between module eigenvalue and clinical variables in the COPDGene (**D**) and SPIROMICS (**E**) cohorts for all subjects, males, and females. Module metabolite assignments are based on the full cohort of profiles. Modules with a negative association were assigned shades of purple and those with a positive association were assigned shades of green based on the 10 log10 FDR or nominal *p* value for COPDGene and SPIROMICS, respectively.

**Figure 2 metabolites-11-00161-f002:**
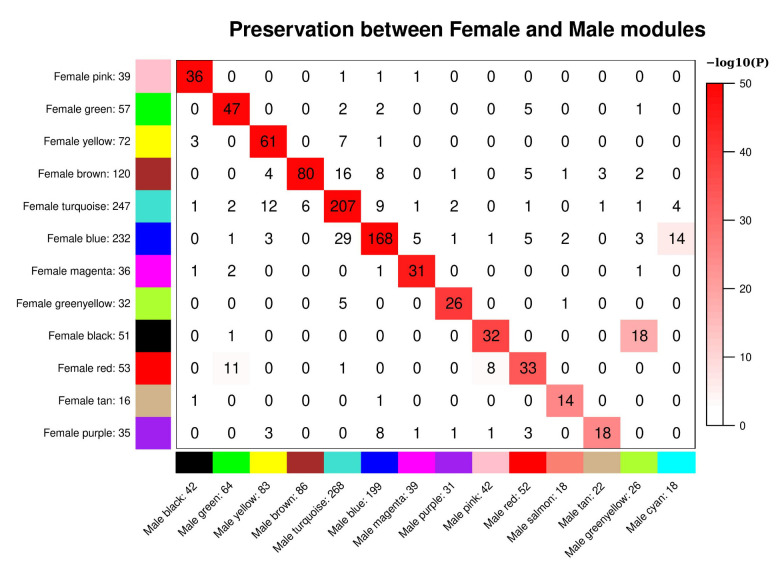
Preservation of female set-specific modules and male set-specific modules in Figure 2. Preservation of female set-specific modules and male set-specific modules in *COPDGene*. Each row of the table corresponds to one male set-specific module (labeled by color as well as text), and each column corresponds to one female set-specific module. Numbers in the table indicate metabolite counts in the intersection of the corresponding modules. Coloring of the table encodes−log(*p*), with *p* being the Fisher’s exact test *p*-value for the overlap of the two modules. The stronger the red color, the more significant the overlap is.

**Figure 3 metabolites-11-00161-f003:**
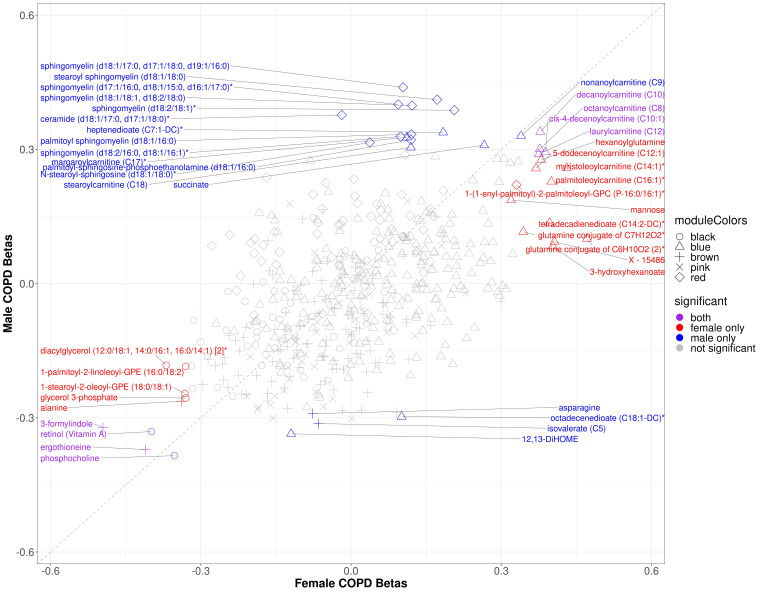
Scatter plot of sex specific betas for COPD models in *COPDGene*. The *x*-axis represents the beta estimates in the female stratum while the *y*-axis represents the beta estimates in the male stratum. Point shape corresponds with module assignment. Points are colored by significance in specific strata.

**Figure 4 metabolites-11-00161-f004:**
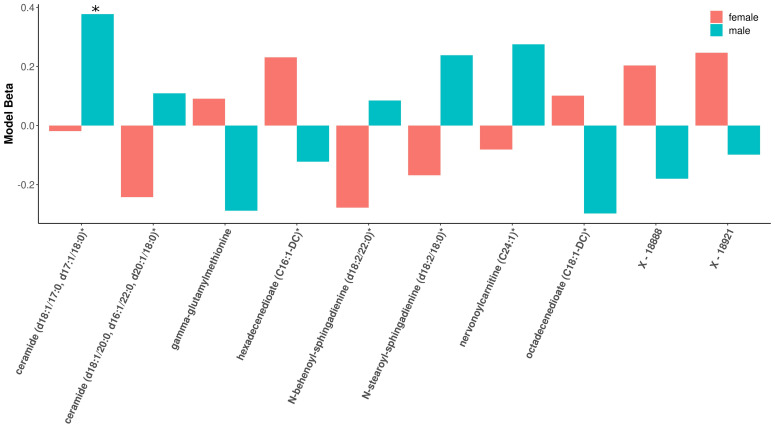
Barplot of beta estimates most divergent by sex. Metabolites along *x*-axis represent the 10 metabolites with the most sex-divergent beta estimates for COPD models. The red bars represent females, while the blue bars are for males. Only ceramide (d18:1/17:0, d17:1/18:0)* reached significance in males.

**Figure 5 metabolites-11-00161-f005:**
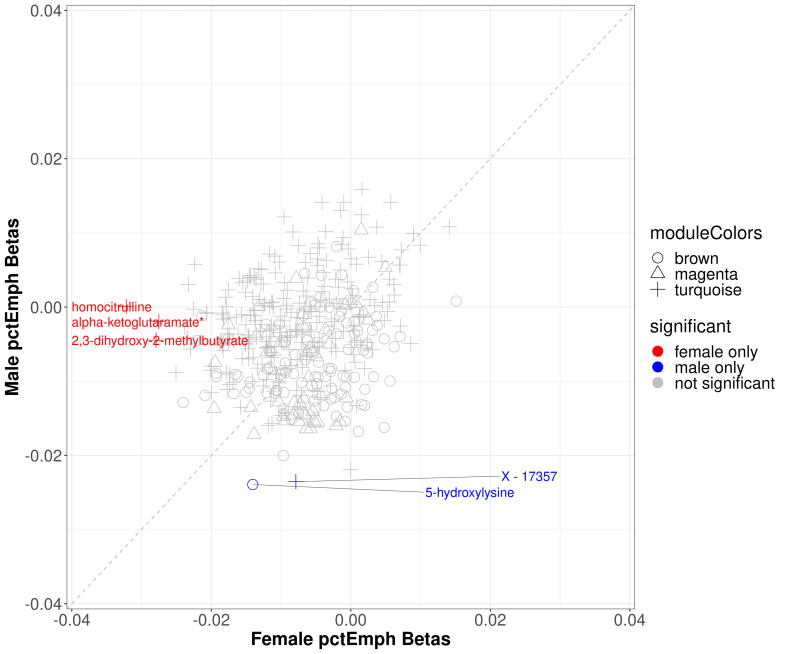
Scatter plot of sex specific betas in percent emphysema. The *x*-axis represents the beta estimates in the female stratum while the *y*-axis represents the beta estimates in the male stratum. Point shape corresponds with module assignment. Points are colored by significance in specific strata.

**Table 1 metabolites-11-00161-t001:** Demographics of Cohorts by Sex.

	*COPDGene*			*SPIROMICS*		
Variable ^a^	Males	Females	*p*-Value ^b^	Males	Females	*p*-Value ^b^
Participants	434	405		232	214	
Age	68.5 (8.4)	66.1 (8.8)	<0.0001	64 (8.0)	63 (8.8)	0.2244
NHW (%)	399 (91.9)	370 (91.4)	0.8593	191 (82.3)	163 (76.2)	0.1365
BMI	29.1 (5.6)	28.6 (6.6)	0.1815	28.6 (4.9)	28.4 (5.6)	0.6088
Current Smokers (%)	88 (20.3)	111 (27.4)	0.0190	90 (39.1)	70 (32.9)	0.2030
Smoking Pack-years	50.1 (27.1)	39.4 (20.5)	<0.0001	55.6 (38.1)	45.5 (21.3)	0.0006
COPD Cases	224 (51.6)	167 (41.2)	0.0033	140 (61.1)	102 (47.7)	0.0193
Percent Emphysema ^c^	9 (11.3)	6.3 (10.2)	0.0005	5.4 (9.1)	5 (8.8)	0.6870

^a^ mean and (standard deviation) reported unless otherwise specified. NHW: non-Hispanic White; BMI: body mass index. ^b^ Chi-square tests were performed for binary variables, *t*-tests were performed for continuous variables. ^c^ In COPDGene, 92.6% of Males and 94.3% of Females had CT data. In SPIROMICS, 87.1% of Males and 84.6% of Females had CT data.

**Table 2 metabolites-11-00161-t002:** Metabolite Classes by module.

*COPDGene* Module	Most Preserved *SPIROMICS* Module	Metabolite Classes *
blue	turquoise	Acyl Carnitines, Fatty Acids (Dicarboxylate, Monohydroxy, Long chain, Medium chain), Endocannabinoids, Nucleotides
red	yellow	Ceramides, Sphingomyelins
turquoise	blue	Xenobiotics, Amino Acids (Tryptophan metabolism, Glutamate metabolism, Histidine metabolism, Branched Chain Amino Acids, Glycine, Serine and Threonine Metabolism, Methionine, Cysteine, SAM and Taurine Metabolism, Polyamine Metabolism, Urea cycle; Arginine and Proline Metabolism), TCA cycle metabolites
brown	brown	Amino Acids (Gamma-glutamyl Amino Acid, Glutamate Metabolism, Branched Chain Amino Acids, Urea cycle; Arginine and Proline Metabolism, Lysine Metabolism, Methionine, Cysteine, SAM and Taurine Metabolism, Phenylalanine Metabolism), Bile Acids, Acyl Cholines, Lysophospholipids
yellow	black	Xenobiotics (Benzoates, Xanthines, Nutritional)
green	green	Lysophospholipids, Phosphatidylcholines (PC), Phosphatidylinositols (PI), Plasmalogens
magenta	pink	Sterioids (Androgenic, Pregnenolone, Corticosteroids, Progestin)
black	purple	Diacylglycerols, Phosphatidylethanolamines (PE), Acyl Carnitines
greenyellow	magenta	Cofactors and Vitamins
purple	NA	Acetylated peptides, Xenobiotics (Benzoates), Secondary Bile Acids
pink	NA	Xenobiotics (Chemicals), Dipeptides, Hemoglobin and Porphyrin Metabolites

* Based on Metabolon’s “Super Pathway” and within class “Sub Pathway” annotation.

**Table 3 metabolites-11-00161-t003:** *COPDGene* and *SPIROMICS* module hub metabolites.

*COPDGene* Module	HubMets	Kme *	*SPIROMICS* Module	HubMets	Kme *
Black	1-stearoyl-2-arachidonoyl-GPE (18:0/20:4)	0.81	Purple	1-stearoyl-2-arachidonoyl-GPE (18:0/20:4)	0.87
Blue	10-nonadecenoate (19:1n9)	0.90	Turquoise	10-nonadecenoate (19:1n9)	0.86
Blue	10-heptadecenoate (17:1n7)	0.87	Turquoise	10-heptadecenoate (17:1n7)	0.87
Blue	oleate/vaccenate (18:1)	0.88	Turquoise	oleate/vaccenate (18:1)	0.89
Brown	gamma-glutamylleucine	0.85	Brown	gamma-glutamylleucine	0.85
Green	1-stearoyl-GPE (18:0)	0.82	Green	1-stearoyl-GPE (18:0)	0.81
Greenyellow	oxalate (ethanedioate)	0.88	Magenta	oxalate (ethanedioate)	0.87
Magenta	androstenediol (3beta,17beta) disulfate (2)	0.92	Pink	androstenediol (3beta,17beta) disulfate (2)	0.888
Pink	X-11442	0.92	None		
Pink	biliverdin	0.76	Turquoise	biliverdin	0.4
Purple	*p*-cresol sulfate	0.89	Blue	*p*-cresol sulfate	0.4
Red	sphingomyelin (d17:2/16:0, d18:2/15:0) *	0.80	Yellow	sphingomyelin (d17:2/16:0, d18:2/15:0) *	0.78
Red	sphingomyelin (d18:2/23:0, d18:1/23:1, d17:1/24:1) *	0.75	Yellow	sphingomyelin (d18:2/23:0, d18:1/23:1, d17:1/24:1) *	0.87
Turquoise	2,3-dihydroxy-5-methylthio-4-pentenoate (DMTPA) *	0.84	Blue	2,3-dihydroxy-5-methylthio-4-pentenoate (DMTPA) *	0.85
Turquoise	pseudouridine	0.84	Blue	pseudouridine	0.8
Yellow	3-hydroxypyridine sulfate	0.88	Black	3-hydroxypyridine sulfate	0.76
Yellow	catechol sulfate	0.87	Black	catechol sulfate	0.8
Yellow	trigonelline (*N*′-methylnicotinate)	0.79	Black	trigonelline (*N*′-methylnicotinate)	0.81

* Kme-correlation coefficient between module 1st principal component and metabolite.

## Data Availability

Clinical data with definitions can be found on dbGaP for COPDGene (phs000179.v6.p2) and SPIROMICS (phs001119.v1.p1). For COPDGene and SPIROMICS, the following clinical data were used: COPDGene_P1P2_All_Visit_29Sep2018 and V5_DERV_INV1_200127, respectively. COPDGene and SPIROMICS metabolomic data are available at the NIH Common Fund’s National Metabolomics Data Repository (NMDR) website, the Metabolomics Workbench, https://www.metabolomicsworkbench.org (accessed on 30 December 2020) (Study IDs ST001443 and ST001639, respectively).

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
