# Peer review of "Metabolomic Profiling Reveals Sex Specific Associations with Chronic Obstructive Pulmonary Disease and Emphysema"

_metabolites, 2021, doi:10.3390/metabo11030161_

Round 1
Reviewer 1 Report
Please see attachment.

Author Response
Gillenwater and coauthors conducted an interesting study on sex specific metabolite associations with COPD and emphysema. In the context of personalized medicine, the topic of this study is certainly interesting. However, I ask the authors to consider the following comments.
-
The same metabolite in COPDgen module and SPIROMICS module is depicted with different colors. To make the reading more immediate it would be better if the colors of each metabolite were the same in the two datasets.
We agree with the reviewer that having the same colors for modules would be conducive for comparison, however, the modules in each cohort are not directly comparable. We performed the WGCNA analysis on both cohorts independently instead of forcing the module structure from COPDGene onto the SPIROMICS metabolites. We did chose experimental design based on the cohorts having slightly different metabolites quantified based on the Metabolon’s Global platform.
However, to make the modules between cohorts more comparable, we have better aligned heatmaps in Figure 1. Figure 1C now shows the preserved modules more clearly based on reorder. Panels 1 D&1E of Figure 1 are now aligned based on the preservation statistics in panel 1C. Thus, comparisons between 1D&1E can be done based on module row, with the exception of the last two modules in both cohorts which showed low preservation in comparison.
-
Although numerous differences, in terms of correlations, are identified in the various groups analyzed, their discussion is very limited. I suggest the authors to deepen the discussion by explaining more thoroughly the biochemistry and the metabolic pathways that are impacted by these differences and why they are relevant in the context of COPD.
We thank the Reviewer for this comment. To further explain the biochemical relationship of increased abundances of acyl carnitines, we have added the following to the results:
“The sex-specific associations for acylcarnitines and COPD may be related to mitochondrial dysfunction. Higher circulating acylcarnitne abundances with increased inflammation and risk of cardiovascular disease (49-52) . Physiologically, acyl carnitines function in mitochondrial metabolism of fatty acids, the removal of excess acyl groups from the body, and the modulation of intracellular coenzyme A homeostasis (52). While acyl carnitines dysregulation affects both sexes, in COPD the effect is more pronounced in females (53–57). Recently, another analysis within SPIROMICS identified positive correlations between urine mitochodrial DNA (mtDNA), an indicator of mitochondrial dysfunction, and respiratory symptoms specifically within females (58). Since mtDNA is maternally inherited, our findings combined with the previous literature support the hypothesis of heritable sex differences in mitochondrial regulation leading female-specific subphenotype within COPD (59).”
-
To better contextualize this study, a brief introduction on serum metabolomics and COPD (reporting the main findings of papers already published, i.e. Wang, et al. PLoS One 2013, ZÄ…bek, et al. Metabolomics 2015, Vignoli et al. J. Proteome Res. 2020) should be presented. Mention should also be made of metabolomics studies on exhaled breath condensate and COPD.
To orient the reader to the current state of understanding of metabolomics analysis in COPD, we have expanded the introduction to include a discussion of previous findings in serum metabolomics. Moreover, we’ve included a discussion on the findings of different biofluids from which metabolic profiles can be derived including exhaled breath condensate.
Reviewer 2 Report
The article is interesting, a representative sample is analyzed, the conclusions deserve attention. In my opinion, Figure 1 is difficult to perceive. It is necessary to make the captions on the figure more readable. It is necessary to add a separate list of the decoding of the designations: pink, purple, etc. Better right in front of Figure 1 and Table 2. Please clarify metabolic profile was determined only in patients with COPD? Are the metabolites discussed in this article associated with COPD? I would like to see comparative data on metabolites in patients with COPD and without, depending on gender. Were metabolic changes assessed depending on the severity of COPD according to the GOLD scale? The conclusion is not highlighted in a separate section.
Author Response
The article is interesting, a representative sample is analyzed, the conclusions deserve attention.
-
In my opinion, Figure 1 is difficult to perceive. It is necessary to make the captions on the figure more readable. It is necessary to add a separate list of the decoding of the designations: pink, purple, etc. Better right in front of Figure 1 and Table 2.
To make Figure 1 more straightforward, we have better aligned heatmaps. Figure 1C now shows the preserved modules more clearly based on reorder. Panels 1D&1E of Figure 1 are now aligned based on the preservation statistics in panel 1C. Thus, comparisons between COPDGene and SPIROMICS WGCNA results (Figure 1D&1E) can be done based on module row, with the exception of the last two modules in both cohorts which showed low preservation in comparison.
Also, we have added another supplementary table (updated S2) that indicates the metabolite classes contained within each module.
-
Please clarify metabolic profile was determined only in patients with COPD? Are the metabolites discussed in this article associated with COPD?
The metabolite profiles were quantified for subjects with and without COPD subjects. All subject metabolite profiles were used to determine the WGCNA modules for each cohort. While metabolites were associated with COPD in the full cohorts (167 in COPDGene and 9 in SPIROMICS at a false discovery rate of 10%), this study sought to elucidate whether individuals of each sex exhibited differential profiles by COPD status.
-
I would like to see comparative data on metabolites in patients with COPD and without, depending on gender. Were metabolic changes assessed depending on the severity of COPD according to the GOLD scale?
We thank the Reviewer for this comment. Figure 3 shows the beta estimates from association tests with COPD status by sex. This scatter plot combines the full cohort and sex-stratified individual metabolite association tests with COPD. To clarify those results a sentence has been added to the caption of Figure 3, as well as Figure 5, which is a similar figure but for percent emphysema.
We assessed metabolic sex-differences using COPD case status and excluded individuals with a GOLD status of 1. We chose this experimental design to increase statistical power in detecting associations between balanced cases and controls (391 vs. 448, respectively) since the group size was increasingly decreased by GOLD severity (0 = 448, 2= 209, 3 = 119, and 4 = 63). As a supplementary analysis, we investigate associations between metabolite profiles in the full cohort and sex-stratified subsets with GOLD status in COPDGene. We found more metabolites to be associated with GOLD status than COPD status (339 vs. 166 in the full cohort, 81 vs 36 in females, and 154 vs. 42 in males). The majority of metabolites associated with COPD were contained within the GOLD associated set for each cohort tested (129/166 in the full cohort, 35/36 in females, and 38/42 in males). This likely demonstrates the heterogeneity among metabolites within GOLD groups and demonstrates the need to deconvolute the metabolite associations by spirometric severity.
-
The conclusion is not highlighted in a separate section.
Thank you for noting this lack of section separation, The concluding paragraph is now highlighted as a separate section.
Reviewer 3 Report
In this report, Gillenwater et al. present an analysis of metabolome profiles in two large cohorts (COPDGene and SPIROMICS) on plasma samples to identify sex specific associations with COPD and emphysema. They resort to an interesting bioinformatics approach of weighted gene co-expression network analysis (WGCNA) that identifies clusters of metabolites distinguishing groups of patients in a given cohort. The whole manuscript is well written and despite the large quantity of data reads fairly easy. The results detailed the main findings and the discussion sufficiently address the issues, perspectives and impact. Nonetheless, I have some concerns that require authors’ attention:
Major comments
- Plasma was used as a source of metabolites but as the authors mentioned in their manuscript and in their previous work ([23]), LBA is a better source of metabolites to identify robust associations. Why the authors did chose plasma instead of LBA for this WGCNA?
- I found rather confusing that the names of metabolites or their families according to each colour module would be mentioned into brackets throughout the manuscript (see for example lines 211-212; 239-241;249-251; etc.). I would recommend to draw a figure or a table providing the main families of metabolites according to their associated modules. It would ideally complement Sup Table S2 and clarify the results.
- I have some issues interpreting the data in a similar way than the authors:
- Figure 1C: module preservation does not seem to be observed for pink and purple because of the small quantity of metabolites but with respectively 12 and 11 matches for the pink and the purple the reader would assume to reach a significant overlap.
- Supp Figure S5: if black and blue female modules are being split over, what about green, brown, and turquoise?
- In general, the authors should give additional clues for non-specialists in WGCNA on data interpretation and guide the reader in the results with mentioning the corresponding panels and tables (see in particular Results 3.3.)
- Supplemental Figure S5 should appear in main figures.
- The authors should discuss more the lack of significant associations observed in SPIROMICS regarding to COPD and emphysema and possibly include the comparative analysis of this second cohort in supplemental data since the main topic of this article as mentioned in the title is the association of metabolomics profiles with COPD and emphysema.
Minor comments
- Regarding to Methods 2.3.: it should be mentioned “see supplemental methods” as it is described in this section. In addition the authors should only specify what is different from what has been published in previous studies concerning metabolites profiling and data processing since they are not providing novel data here but a complementary analysis of publically available data.
- Fig1C, provide legend on the figure for the colour scale (-log(p))
- Line 222: precise the Figure we are looking at?
- Lines 248-288: precise the figures we are looking at for each section?
- What is the distinction between “correspondence” and “preservation”? The authors seemed to use them as synonyms (see Figure 1C/SF5-8).
- Figure 2: for clarity, module colours should keep the colour code and significant labels may take different shapes.
- Figure 3: keep axis as Figure 2 (or possibly use a range of -0.5/+0.5 for both figures. In addition, add the statistics sign on the appropriate bar.
- Lines 391-393: please add the % of unknown annotation.
- Lines 398-399: it is confusing since the association with COPD does not concern SPIROMICS.
- Lines 401-403: this is also confusing since as evidenced in Figure 1D, the difference between males and females in terms of smoking is very discreet.
Author Response
In this report, Gillenwater et al. present an analysis of metabolome profiles in two large cohorts (COPDGene and SPIROMICS) on plasma samples to identify sex specific associations with COPD and emphysema. They resort to an interesting bioinformatics approach of weighted gene co-expression network analysis (WGCNA) that identifies clusters of metabolites distinguishing groups of patients in a given cohort. The whole manuscript is well written and despite the large quantity of data reads fairly easy. The results detailed the main findings and the discussion sufficiently address the issues, perspectives and impact. Nonetheless, I have some concerns that require authors’ attention:
Major comments
-
Plasma was used as a source of metabolites but as the authors mentioned in their manuscript and in their previous work ([23]), LBA is a better source of metabolites to identify robust associations. Why the authors did chose plasma instead of LBA for this WGCNA?
The collection of LBA is invasive and not suitable for large scale population-based studies.
COPD has systemic features and the plasma metabolome likely captures some of these features while simultaneously affording the analysis of large numbers of subjects stratified by sex.
Russ, could you comment on this?
-
I found rather confusing that the names of metabolites or their families according to each colour module would be mentioned into brackets throughout the manuscript (see for example lines 211-212; 239-241;249-251; etc.). I would recommend to draw a figure or a table providing the main families of metabolites according to their associated modules. It would ideally complement Sup Table S2 and clarify the results.
Thank you for noting this. We have clarified the metabolite names/families in the updated Supplementary Table 2, which contains the breakdown of metabolite classes by module. This should better serve as a map for module associations outlined in Figure 1 and complements the results of Supplementary tables 2 and 3.
-
I have some issues interpreting the data in a similar way than the authors:
-
Figure 1C: module preservation does not seem to be observed for pink and purple because of the small quantity of metabolites but with respectively 12 and 11 matches for the pink and the purple the reader would assume to reach a significant overlap.
-
We appreciate this important point by the Reviewer. Both of these modules have nominally significant preservation (Fisher’s exact p values of 0.02 and 0.04 for COPDGene modules purple and pink, respectively)(See table S5 for module preservation statistics). However, since the COPDGene modules are subsumed by the larger SPIROMICS modules, the structure of these modules is not preserved in the other cohort.
-
Supp Figure S5: if black and blue female modules are being split over, what about green, brown, and turquoise?
We thank the Reviewer for this insightful observation. Upon further inspection the metabolites from the black, blue, brown, turquoise, and red female modules have groups of at least 10 metabolites that are not preserved between sexes. This has been addressed the results.
-
In general, the authors should give additional clues for non-specialists in WGCNA on data interpretation and guide the reader in the results with mentioning the corresponding panels and tables (see in particular Results 3.3.)
We thank the Reviewer for this recommendation. To clarify what is obtained from the WGCNA modules, a paragraph has been added to the beginning of Results section 3.3. This prepare the reader for interpretation of the remaining results in this section. Also, corresponding figure panels are now referenced throughout this section to guide the reader to appropriate visualizations.
-
Supplemental Figure S5 should appear in main figures.
We have updated the main figures and this is now Figure 2 in the main figures.
-
The authors should discuss more the lack of significant associations observed in SPIROMICS regarding to COPD and emphysema and possibly include the comparative analysis of this second cohort in supplemental data since the main topic of this article as mentioned in the title is the association of metabolomics profiles with COPD and emphysema.
We thank the reviewer for this comment, as it is important for interpretation of our findings. To the results section in the manuscript, we have further described the association results in SPIROMICS at a nominal p-value. While not reaching significance with multiple comparison corrections (potentially due to the much smaller sample size in SPIROMICS), this demonstrates where similar relationships exists within the metabolic profiles of SPIROMICS subjects. The demographic differences between the two cohorts are outlined in the results section and to the discussion we related that diet and pharmaceutical differences were not taken into account.
Minor comments
-
Regarding to Methods 2.3.: it should be mentioned “see supplemental methods” as it is described in this section. In addition the authors should only specify what is different from what has been published in previous studies concerning metabolites profiling and data processing since they are not providing novel data here but a complementary analysis of publically available data.
Thank you for this comment. As the metabolomic data were not processed further before applying the exclusion criteria to the cohort, reference is now only made to the complementary analysis and the supplemental methods have been removed.
-
Fig1C, provide legend on the figure for the colour scale (-log(p))
Thank you. A legend has been added.
-
Line 222: precise the Figure we are looking at?
This statement is referring to Figure 1C. The has been updated in the text.
-
Lines 248-288: precise the figures we are looking at for each section?
This statement is referring to Figure 1E. This has been updated in the text.
-
What is the distinction between “correspondence” and “preservation”? The authors seemed to use them as synonyms (see Figure 1C/SF5-8).
There is no difference. For clarity, we have changed “correspondence” to “preservation” in the text and figures.
-
Figure 2: for clarity, module colours should keep the colour code and significant labels may take different shapes.
We thank the Reviewer for this comment. This was our thought at first, as well. However, we found it very difficult to distinguish the metabolites associated in a sex-specific manner from those associated in both sexes due to the overlapping points. Thus, we decided to use colors based on sex to emphasize the differences.
-
Figure 3: keep axis as Figure 2 (or possibly use a range of -0.5/+0.5 for both figures. In addition, add the statistics sign on the appropriate bar.
We thank the Reviewer for this comment. The scale has been updated and the significance is now indicated.
-
Lines 391-393: please add the % of unknown annotation.
Thank you, this has been updated in the text.
-
Lines 398-399: it is confusing since the association with COPD does not concern SPIROMICS.
We thank the reviewer for this comment, we have revised this sentence for clarity. “ Second, while the COPDGene and SPIROMICS cohorts are two large, well-characterized COPD cohorts, pharmaceutical data and diet differences were not considered.”
-
Lines 401-403: this is also confusing since as evidenced in Figure 1D, the difference between males and females in terms of smoking is very discreet.
We thank the reviewer for pointing out the confusion in this statement. The intention was to review how Metabolon continues to update the metabolites they quantify. Since the data were collected years apart, the metabolite sets quantified were not an exact match. This is now addressed in the third limitation.
Reviewer 4 Report
Metabolomics measures hundreds of endogenous and exogenous (xenobiotic) small molecules simultaneously, providing an opportunity to identify biomarkers of exposure to cigarette smoke and markers that reflect host-related metabolic adaptations. These markers as analysed in the paper may provide new insights into the biological mechanisms of addiction and smoking-related diseases.
Minor point
As same as you analyse as individual associations age and BMI in addition to emphysema or sex, you should do the same with smoking intensity (pack-years), not only adjusting for it in the linear regression model.
Add a new paragraph about it
Author Response
Metabolomics measures hundreds of endogenous and exogenous (xenobiotic) small molecules simultaneously, providing an opportunity to identify biomarkers of exposure to cigarette smoke and markers that reflect host-related metabolic adaptations. These markers as analysed in the paper may provide new insights into the biological mechanisms of addiction and smoking-related diseases.
Minor point
As same as you analyse as individual associations age and BMI in addition to emphysema or sex, you should do the same with smoking intensity (pack-years), not only adjusting for it in the linear regression model.
As the smoking phenotype is of great interest to pulmonary disease, we did test for associations with smoking intensity. Please refer to figure 1D and 1E. This is also referenced in the results section on co-variate associations.
Round 2
Reviewer 1 Report
The authors satisfactorily addressed all the comments emerged during the previous step of peer-review. There are no other comments from my side.
Reviewer 3 Report
The authors answered all my comments and significantly improved their manuscript in this R1. I have no additionnal comments.